# Numerical and Experimental Investigation of Quasi-Static Crushing Behaviors of Steel Tubular Structures

**DOI:** 10.3390/ma15062107

**Published:** 2022-03-12

**Authors:** Young IL Park, Jin-Seong Cho, Jeong-Hwan Kim

**Affiliations:** Department of Naval Architecture and Offshore Engineering, Dong-A University, Busan 49315, Korea; parkyi1973@dau.ac.kr (Y.I.P.); 1532897@donga.ac.kr (J.-S.C.)

**Keywords:** quasi-static crushing test, steel tubular structure, nonlinear finite element simulations

## Abstract

In this study, a numerical and experimental investigation of the quasi-static crushing behavior of steel tubular structures was conducted. As the crushing failure behavior involves a high level of nonlinearity for the numerical simulations, these were compared with previous experimental works, including crushing tests of steel square tubes to calibrate the numerical results. Six parameters for the numerical simulations, namely (1) loading boundary condition, (2) geometrical imperfection, (3) friction coefficient, (4) element size, (5) element type, and (6) material nonlinearity model, were examined using a series of finite element analyses. Through the sensitivity study for each parameter, the deformation and crushing load of the steel tube were investigated, and the value that best matched the experimental results was selected. The results of the numerical analysis for the determined model were compared with the experimental results. Finally, the authors provided recommendations that should be considered when performing nonlinear finite element simulations of crushing failure events.

## 1. Introduction

Most of the world’s cargo volume is transported using ships, and the volume of ship traffic continues to increase. The growing shipping volume results in high maritime traffic, resulting in various accidents. Among these accidents, ship collisions occur most frequently [1]. Therefore, it is necessary to establish countermeasures for large- and small-scale vessel collision accidents.

A ship is a plated structure consisting of a stiffened panel, and buckling or crushing of the structural members can occur due to an excessive external force, such as that arising from a collision of the hull structure during operation. In severe cases, tearing and fracture may occur, resulting in personal damage and environmental pollution. In the past several decades, many experimental, analytical, and numerical studies have been conducted to elucidate the mechanism of damage caused by the collision of structures.

Viswabrahmana and Suresh [2] carried out experimental studies on tailor-welded blank applications in automotive vehicles for frontal crash validation, using a three-point tubular bending specimen. Ataabadi et al. [3] numerically examined the dynamic axial crushing response of cylindrical tubes made of unidirectional carbon fiber-epoxy materials. Dong and Fan [4] developed an analytical technique to predict the mean crushing load for hybrid square tubes, consisting of metal and composite materials. Paik et al. [5] carried out numerical and experimental evaluations of the crushing behaviors of ASTM A500 type carbon steel under low-temperature conditions. Paik et al. [6] experimentally and theoretically examined an aluminum honeycomb sandwich panel, with respect to its bending and crushing capacities. Wang and Suzuki [7] derived a simplified calculation formula for the crushing strength of the bow structure of a ship in a bow collision event. Paik and Pedersen [8] presented an analysis method for the residual strength of the side structure of a double-hull tanker after a ship–ship collision event. Paik and Pedersen [9] studied the damage inflicted to hull structures in ship collisions using the idealized structural unit method. Ohtsubo and Suzuki [10] presented an accurate calculation method for the mean crushing strength of a ship’s bow structure and proposed a formula design by comparing the results of finite element analyses (FEAs) with the existing experimental works. Jones and Birch [11] conducted experimental studies on the structural behavior of square section steel tubes, with various heights and numbers of stiffeners, against axial impact loads. Pugsley [12] carried out impact tests on longitudinally stiffened steel tubes to investigate the crushing behavior of a railway vehicle for collision cases and formulated the mean crushing load based on the stiffness ratio. Macaulay and Redwood [13] conducted impact experimental tests for rods and steel tubes, in order to investigate the characteristics of impact load behavior for railway vehicle applications. Lowe et al. [14] found that the dynamic crushing load is higher than the static crushing load, based on the experimental test results for car models. Nagasawa et al. [15,16,17] carried out a quasi-static crushing test for hull and bridge models, in order to estimate the impact load and internal absorbed energy for the case of a collision accident between a ship and a bridge. They evaluated the effectiveness of the shock and energy absorption by model tests of the shock absorber at the fore side, as well as the parallel and corner parts of the test bridge. Ohnishi et al. [18] carried out a crushing test for models of crude oil carriers and container vessels and estimated the amount of damage sustained in a ship collision by calculating the ultimate strength based on FEA. Lee [19] performed experimental studies on the ultimate strength and absorbed energy of thin plating, assuming that the damage caused by the ship’s collision was related to the amount of kinetic energy loss. The relationship between the amount of damage of the plate member and absorbed energy was proposed through experimental and theoretical assessment results. Lee [20] developed a formula design to calculate the absorbed energy for isotropic and orthotropic plates, through theoretical and experimental works. Meng et al. [21] studied the deformation mechanism for square section steel tubes subjected to axial loads by theoretical and experimental studies. They found that the folding mode of the square section steel tubes was constant for the ratio of thickness-to-breadth. Wierzbicki [22] proposed a method to calculate the crushing load for L, T, Y, and X sections, assuming that the plate structures are rigid plastic materials. Amdahl [23] derived a design average crushing strength, with the assumption that hulls consist of L-, T-, and X-shaped structural elements. In addition, the applicability of the proposed formula was evaluated through experimental works for a ship bow model. Abramowicz [24] performed a theoretical analysis of the effective crushing distance and proposed a design equation for the mean crushing strength of unstiffened and stiffened square section steel tubes. Abramowicz and Jones [25] derived the dynamic crushing strength using experimental tests for square section steel tubes using drop objects. Kawai et al. [26] derived the collision energy by mixing the finite element (FE) method and rigid-plastic theory, and the energy amount was confirmed experimentally. Yang and Caldwell [27] proposed a design equation to estimate the amount of damage for the collision of a ship’s bow structure and carried out an experimental work to verify the accuracy of the design equation. Toi et al. [28] carried out a crushing test for a square section steel tube and compared the test results with the existing theoretical solutions and empirical formulas.

The structural safety assessment methods used in the above previous studies can be mainly classified into two types: (1) evaluation by the ultimate strength concept and (2) assessment by the amount of absorbed collision energy. The former method is mainly used for normal loading conditions, whereas the latter method is normally used to evaluate the safety of a structural member under collision situations. To secure structural safety against collision accidents between ships and offshore structures, it is critical to understand the crashworthiness of the ship’s structural members in the basic design phase. As the different collision strengths for hull designs, based on different owner requirements, cannot be determined only by experimental and/or theoretical methods, numerical simulations are essential.

While many studies have been carried out on the axial crushing of steel sections, there are few studies that provide guidelines for efficient and accurate numerical simulations. Therefore, it is worthwhile to define the critical parameters affecting the numerical analysis of steel crushing problems and study their influence.

The major aim of the present study is to introduce an efficient and accurate numerical simulation method for the analysis of the structural crashworthiness in ship and offshore production applications. As the crushing failure simulations require one of the most complicated nonlinear techniques, the effect of FE input parameters should be quite critical to crashworthiness behaviors. Thus, in this study, a series of nonlinear FE simulations of crushing failure events were carried out to investigate key input parameters, such as (1) loading boundary conditions, (2) geometrical imperfection, (3) friction coefficient, (4) element size, (5) element type, and (6) material nonlinearity models. The contribution of each parameter to the simulation results was determined and compared with the experimental results. Finally, the authors provided recommendations that should be considered when performing nonlinear FE simulations of crushing failure events.

## 2. Experimental Works

Jung [29] carried out a series of quasi-static crushing tests for thin-walled square tube specimens with axial and/or circumferential stiffeners, including unstiffened specimens, as shown in Figure 1 for the test setup. Jung [29] provided test raw data for the current work tasks, as described in this section.

In this study, the experimental results of unstiffened, thin-walled, square tube specimens, with six different dimensions, are listed in Table 1, and the specimen shape and loading conditions are shown in Figure 2. The specimens were made of ASTM A500 carbon steel (Grade A), and the mechanical properties are given in Table 2. The specimens were quasi-statically crushed at a speed of 0.05 mm/s, which was determined considering the loading types, according to the applied strain rate, defined by ASME [30].

Figure 3 shows typical crushing response of an unstiffened specimen (denoted as US), and Figure 4 shows the corresponding deformed shape of the specimen. The characteristics of the deformed phase are as follows:

OA phase: This phase represents the load up to the compressive collapse of the specimen; before reaching point A, the specimen begins to buckle and leads to the ultimate limit phase.

AB phase: After reaching point A (ultimate strength), the crushing deformation further increases, internal force decreases along A–B, and first folding occurs near point B, slowing the deterioration of the internal force.

BC phase: After the first folding event is finished, the internal force rises rapidly, owing to the contact reaction force by the adjacent part, and it continues until the second folding occurs at point C. The direction of the second folding occurred in the opposite direction to the first folding.

CD phase: The second folding occurs at point C, and the internal force decreases until point D, where the folding is completed.

DE phase: Above, the deformation behavior appears repeatedly until folding no longer occurs.

Point E: Folding no longer occurs, and the specimen behaves as a rigid body.

The absorbed energy of the sample was calculated by integrating the reaction force–indentation curve with the maximum deformation. Therefore, the mean axial crushing force (P_m_) in Figure 3 was obtained as the absorbed energy, divided by the maximum deformation.

Figure 5 shows the experimental results of six specimens (US-1 to US-6), as a relationship between the indentation and reaction force. It can be observed that the reaction force of US-4, which has the largest t and b, is the largest. As mentioned in the introduction, this study conducted a parametric study and developed the simulation model that best represented the experimental results.

## 3. Finite Element Analysis

### 3.1. Specifications of the Finite Element Analyses

As the crushing failure behavior is significantly complicated, owing to the surface–surface contact and folding behavior, the nonlinear FE assessment results should be carefully calibrated by the experimental results, in order to determine the nonlinear material models, “surface to surface contact” properties, boundary conditions, etc.

Nonlinear static FEAs were conducted for the following parameters to investigate the nonlinear behavior related to crushing failure:(1)Boundary condition;(2)Geometrical imperfection;(3)Friction coefficient;(4)Material nonlinearity model (perfect elasto-plastic model vs. multilinear plastic model);(5)Element size;(6)Element type (full integration vs. reduced integration shell).

For the FE simulations, the commercial FE code ABAQUS was employed, and the specifications are listed in Table 3.

### 3.2. Crushing Length

The mean crushing force was obtained using the absorbed energy and maximum deformation, which is the crushing length. The crushing length can be defined as the total indentation length, until the specimen becomes a rigid object, as shown in Figure 6c of Figure 6. However, in contrast to experimental measurements, the maximum indentation does not clearly appear for the FE simulation cases, and it is often difficult to define the crushing length visually. Therefore, a criterion is required to define the crushing length in the FE simulation.

We plotted the sum of the contact forces by folding together with the reaction force, as shown in Figure 7. It was observed that the contact force increased rapidly at a certain point, where folding was completed. Therefore, in this study, the crushing length in the FE simulations was determined as that just before the point where the crushing contact pressure rapidly increased.

### 3.3. Parameter #1: Boundary Condition

The loading condition of the experimental work is a simple contact between the loading surface of the crosshead and top of the test specimen. Figure 8 shows a schematic of the theoretical loading conditions, such as the fixed loading (left figure) and contact boundary conditions (right figure). The figure also shows the difference in deflection patterns, due to these boundary conditions, especially regarding the folding pattern.

The boundary condition of the experimental works could be similar to the contact boundary condition, but the actual loading condition may contain a certain level of rotational restraint, owing to the friction effect. Thus, the actual boundary condition could be in the middle of the fixed and contact boundary conditions.

In this study, a series of FEAs on the effect of these boundary conditions on the crushing strength behavior were carried out and compared with the experimental test results.

Figure 9 and Figure 10 show the effect of the loading boundary conditions on the crushing behavior of the reaction force and internal energy for the US-1 specimen, respectively.

Table 4 summarizes the simulation results and compares them with the experimental results for ultimate (P_u_) and mean crushing loads (P_m_). The results show that 27.6% and 31.7% are the maximum differences between the test results and FE simulations for the free and fixed boundary conditions, respectively, regarding the ultimate crushing loads.

Regarding the mean crushing loads, 29.3% and 23.8% are the maximum differences between the test results and FE simulations for free and fixed boundary conditions, respectively.

The FE simulations show a more conservative tendency than the experimental results. Further, the results with fixed boundary conditions show better agreement with the experimental results. However, the application of a free boundary condition could be recommended for use in nonlinear FE simulations, for the purpose of safe design.

### 3.4. Parameter #2: Initial Imperfection

The structural crushing capacities of the specimen should depend on the initial deflection (imperfection) levels; however, no initial imperfection measurements for the tube samples were carried out. Therefore, in the current study, four initial imperfections were applied using eigenmodes with maximum imperfection levels to determine the effect of initial imperfection. The maximum imperfection level (w_0_) was calculated using Equation (1). Three levels of imperfection amplitude (amp) were considered in this study, i.e., 0.05: slight level; 0.10: average level; 0.20 and 0.30: severe level. Figure 11 shows the three eigenmodes of the current test specimen, obtained from linear buckling analyses. In this study, we used the first eigenmode to implement the initial imperfection.
(1)$$w_{0}=\mathrm{amp}\cdot \beta^{2}\cdot t$$
where $\beta =\frac{b}{t}\sqrt{\frac{\sigma_{Y}}{E}}$, $\sigma_{Y}$ = yield strength,
E = elastic modulus, and amp = 0.05, 0.10, 0.20, 0.30.


Figure 12 and Figure 13 show the effects of the initial imperfection level on the crushing behavior of the reaction force and internal energy for the US-1 specimen, respectively.

Table 5 summarizes the crushing FE simulation results for the four initial deflection levels and compares them with the experimental results. The differences between the experimental works and FE simulations are 1–37% and 1–24% for the ultimate (P_u_) and mean crushing strengths (P_m_), respectively.

The minimum difference of mean crushing load between the experimental work and FE simulation came from US-1 with 0.20 of amp of initial imperfection.

The FE simulation results for the other cases, i.e., US-2–US-6, with 0.20, 0.30, 0.30, 0.10, and 0.20 of amp of initial imperfection, respectively, show minimum differences of mean crushing load with the experimental results.

Because the state of the raw plate, with which the test specimen is manufactured, could be different, and the environment may also be different, during the production process, the six specimens may have different initial deflection and residual stress levels. Thus, an accurate mean crushing load can be calculated by FEA, based on accurate initial imperfection measurements.

As the initial imperfection value increased by six times (0.05–0.30), the ultimate crushing strength decreased by approximately 30%, while the mean crushing load decreased by approximately 10%. Thus, it was confirmed that the mean crushing load was not very sensitive to the initial deflection, compared to the ultimate crushing strength.

### 3.5. Parameter #3: Friction Coefficient

During the crushing failure event, the specimen buckled and folding occurred as the crushing deformation increased. After the first folding event, the internal force increased again, and the contact reaction force increased, until the specimen became a rigid object. Owing to the folding event, self-contact nonlinear boundary conditions should be applied to the FE model, and the friction coefficient is one of the required input parameters.

The friction coefficient can be calculated experimentally. However, as there are no test results available for this crushing specimen, sensitivity analyses on the effect of friction coefficient values on the crushing behavior were carried out using four different friction coefficients: 0.04, 0.08, 0.16, and 0.26.

The ultimate crushing loads obtained with the four different friction coefficients were found to be identical to each other, but the mean crushing loads were slightly different. This is thought to be because, for the target specimen, the ultimate crushing load is measured at the first peak, while the mean crushing load is calculated by the total cumulated internal energy when the final folding is finished. Because the friction coefficient is involved in the friction behavior between each folding, it is expected that the greater the number of folding, the greater the effect of friction.

Figure 14 and Figure 15 show the effects of friction coefficients on the crushing behavior of the reaction force and internal energy for the US-1 specimen, respectively. As shown in the figures, the effect of the friction coefficient is almost negligible.

Table 6 summarizes the simulation results for the four different friction coefficients and compares them with the experimental results. The differences between the experimental works and FE simulations are of 4–31.7% and 4.2–24.9% for the ultimate (P_u_) and mean crushing strengths (P_m_), respectively.

As the friction value increased by six times (0.04–0.24), the ultimate crushing strength did not change, while the mean crushing load varied by approximately 5% as a maximum; thus, it was confirmed that the mean crushing load is not sensitive to the friction coefficient.

### 3.6. Parameter #4: Material Property Model

The crushing failure behavior should depend on the nonlinear material model. To investigate the structural nonlinear behavior by nonlinear material properties, two nonlinear models were considered for the FE simulations: (1) perfect elasto-plastic and (2) multi-linear materials, as shown in Figure 16.

For the multi-linear material model, the strain hardening effect is included, which is calculated from tensile coupon test results. For simplicity, it is made so that the stress remains constant after the strain reaches the value of strain-to-fracture. The values required for each material description are shown in Table 2.

Figure 17 and Figure 18 show the effects of nonlinear material models on the crushing behavior of the reaction force and internal energy for US-1 specimen, respectively.

Table 7 compares the simulation results for the two different nonlinear material models with the experimental results. The differences between the experimental works and FE simulations, using the perfect elasto-plastic model, are of 4.0–31.7% and 2.5–31.7% for the ultimate and mean crushing strengths, respectively.

The differences between the experimental works and FE simulations, using the multi-linear material model, are of 3.9–31.0% and 0.4–31.0% for the ultimate and mean crushing strengths, respectively.

Figure 19 shows the final deformed shape and crushing distance, as a result of using each material model. There is not much difference in the number of folding and folding shape, but the case using perfect elasto-plastic model shows a longer crushing distance. However, it can be seen that there is little difference between the two, in terms of absorbing energy.

Therefore, the FE simulations, using the multi-linear nonlinear model, showed slightly better agreement with the experimental results; however, a few FE simulation cases, using the multi-linear nonlinear model, showed higher strength values than the experimental works.

### 3.7. Parameter #5: Finite Element Mesh Size

Element size is one of the most critical parameters that determines the nonlinear results. In particular, in the steel crushing problem, special attention to the element size is required because it significantly affects the folding shape and number of lobes. We performed a sensitivity study, using seven different element sizes, namely 2, 3, 4, 5, 6, 7, and 8 mm, for the US-1 case with a plate thickness of 2.2 mm. Table 8 presents the deformed shape and corresponding number of elements for each folding in each FE model. As can be observed, the smaller the element size is, the smoother and more natural the shape of the lobe. To determine the proper element density, the mean crushing loads for each case are compared in Figure 20. As a result, the mean crushing loads become smaller as the element size decreases, and they converge when the element size is 5 mm.

Figure 21 and Figure 22 show the effects of element size on the crushing behavior of the reaction force and internal energy for the US-1 specimen, respectively, and Table 9 compares the simulation results for six different element sizes with the experimental results.

The difference between the experimental works and FE simulations for the ultimate crushing strength is approximately 10%, whereas it ranges between 2.7% and 14.6% for the mean crushing load.

### 3.8. Parameter #6: Element Type (Reduced Integration vs. Full Integration)

As another modeling parameter for FEAs, the effect of the element type on the shell element was investigated.

In this study, a four-node shell element was applied to the FE simulation with full integration and reduced integration options (S4, S4R). Full integration elements use two integration points in each direction, whereas reduced integration elements have a single integration point at the element centroid [31]. Figure 23 and Figure 24 show the crushing reaction force and internal energy for US-1, respectively.

Table 10 compares the simulation results for different element types with the experimental results.

The FE simulation results of the full integration and reduced integration shell element cases show very similar results for the ultimate crushing strength and mean crushing load, with a maximum difference of 5.5%. Therefore, the reduced integration shell element-based FE model can provide reliable simulation results.

The maximum difference between the experimental works and FE simulations for the ultimate crushing strength was approximately 32%, and it was 25% for the mean crushing load.

## 4. Discussion

### 4.1. Determination of the Parameters for Finite Element Simulation

For the efficient and accurate numerical implementation of the steel tubular structure subjected to axial crushing, the main parameters required for the FE model were determined by a series of case studies. The results of the sensitivity studies on boundary conditions, geometrical imperfections, friction coefficients, material nonlinearity models, and element size and type, using six unstiffened specimens, are presented in Figure 25 results were compared based on the mean crushing force, which indicates the energy absorption power of the tube. The differences between the experimental and analysis results of the six specimens, for the values applied in each parameter, are presented using the box-and-whisker plot [32], except for the element size case, where the difference is presented using a bar chart because only one sample was tested. A box-and-whisker plot is a method for graphically presenting data groups using quartiles. It is a good way to identify errors because it can efficiently express the variation, degree of dispersion, and skewness of each data group. Figure 26 shows the box-and-whisker plot using the relation with the normal distribution. In Figure 25, the quartiles, including the maximum, minimum, and median of each data group, as well as the mean and standard deviation (σ), can be identified. Outliers are statistically insignificant, as they represent data points outside the maximum (+49.65%) and minimum (−49.65%) specified in the box-and-whisker plot. The parameters, determined based on the difference distributions, are listed in Table 11. For each parameter, we selected the case with the least median difference from the comparison between the simulation and experimental results. If the variation in medians between the cases is small, as in the case of initial imperfection, the variance (i.e., interquartile range) was also considered. Unlike these parameters, the element size was determined based on the results of the mesh convergence test in Figure 20, as described in Section 3.7.

### 4.2. Comparison between Numerical Simulation and Experiments

The simulation results, obtained by applying the parameters determined in Table 11, were compared with the experimental results, as shown in Figure 27. The boundary condition at the area where the force is applied is slightly different from the actual condition, so the deformation at the top of the tube is slightly different. However, the number of lobes at each stage and overall shape of the deformation showed good agreement. Figure 28 presents a comparison of the deformed shapes of the specimen for which the test was completed. It can be observed that the shape of the lobe was realistically well implemented in the numerical simulation.

In this study, the mean crushing force was used as a measure for an efficient and accurate numerical simulation because the ability to absorb energy is the most important design factor for a tube subjected to axial crushing [33]. Prior to the comparison of the mean crushing force, the crushing length, which is very important for the calculation of the mean crushing force by numerical simulation, is compared in Figure 29. Unlike the experiment, it is difficult to obtain the crushing length in the numerical simulation; therefore, in this study, the crushing length was determined based on the point at which the contact force suddenly increased. It can be observed that the simulated crushing length agrees well with the experimental results. In Figure 30, the experimental and simulated mean crushing forces are compared. It shows very good agreement, except for the US-4 case, which has the thickest tube plate. US-4 is 30–40% thicker than the other specimens, and further research is recommended to determine the exact cause of the error.

## 5. Conclusions

In this study, the required input parameters for a nonlinear FEA of the crushing failure of steel specimens were categorized, and a sensitivity analysis was carried out, using experimental results.

The parameters determined through the sensitivity analysis are presented in Table 11 and they can be summarized as follows.

The fixed loading boundary condition shows better agreement with the experimental results than the contact loading boundary condition; however, FE assessment, with a fixed boundary condition, could lead to excessive ultimate crushing strength and/or mean crushing loads. Therefore, when a more conservative application is required, for design purposes, a contact loading boundary condition is recommended.

The level of initial imperfection has a particularly large impact on the ultimate crushing strength, but it has a relatively smaller effect on the mean crushing load. However, precise management, related to the initial imperfection, is required for accurate crushing performance estimation. The initial imperfection of the residual stress, caused by the welding process, was excluded in this study, but it requires further analysis.

The effect of the friction coefficient on the crushing capacity was determined to be practically negligible, with a maximum of less than 5%.

The crushing FE assessments by nonlinear material properties, using the tensile coupon test results, showed better agreement with the experimental results; however, the application of perfect elasto-plastic nonlinear material properties is recommended because of practical engineering reasons.

The FE mesh size was found to be a major parameter, affecting both the ultimate crushing strength, as well as the mean crushing load. Therefore, the mesh size for the FE assessment should be determined based on the results of a mesh convergence study.

Finally, the results for the full integration shell element and reduced integration shell element did not differ significantly from each other; therefore, the reduced integration shell element-based FE mesh could be acceptable, in terms of simulation efficiency.

The simulation results, applying the determined parameters, showed very good agreement with the experimental results, in terms of the deformed shape, crushing length, and mean crushing force. Based on the results of this study, it is expected that better results will be obtained if the study is extended to the dynamic crushing test and stiffened tube. In addition, studies considering manufacturing conditions, such as welding and joints, are recommended as further studies.

## Figures and Tables

**Figure 1 materials-15-02107-f001:**
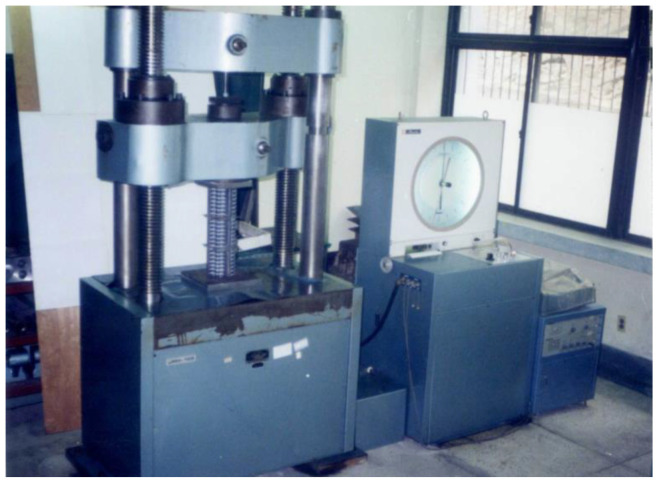
Test setup for crushing test [29].

**Figure 2 materials-15-02107-f002:**
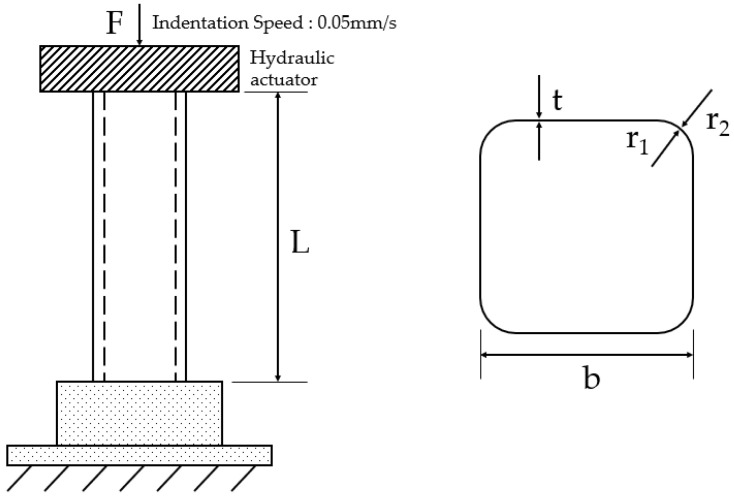
Load condition and specimen shape adapted from [29].

**Figure 3 materials-15-02107-f003:**
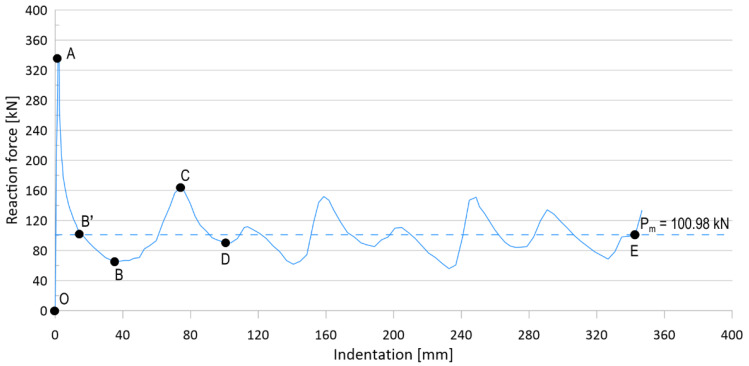
A typical crushing response of an unstiffened specimen.

**Figure 4 materials-15-02107-f004:**
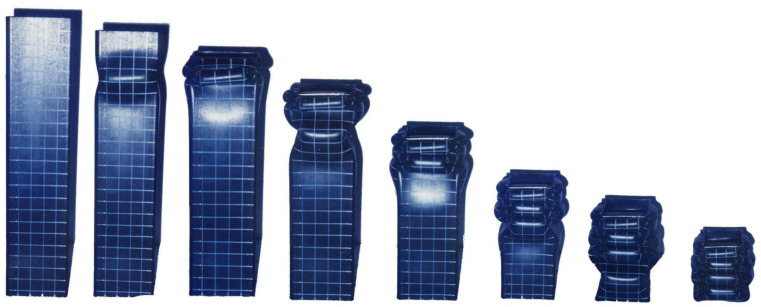
Crushing test of an unstiffened specimen in the experimental work (US-1).

**Figure 5 materials-15-02107-f005:**
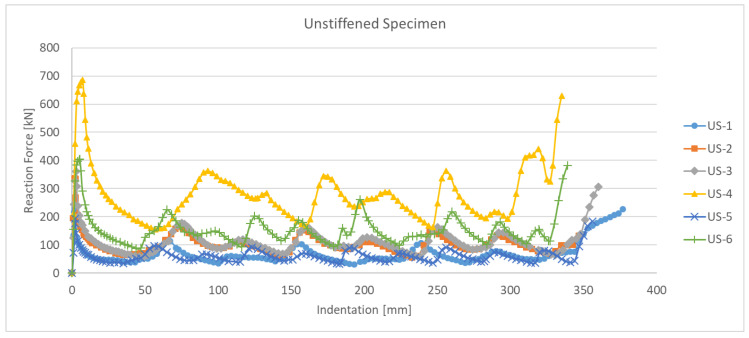
Indentation–reaction force in the crushing test of unstiffened specimens.

**Figure 6 materials-15-02107-f006:**
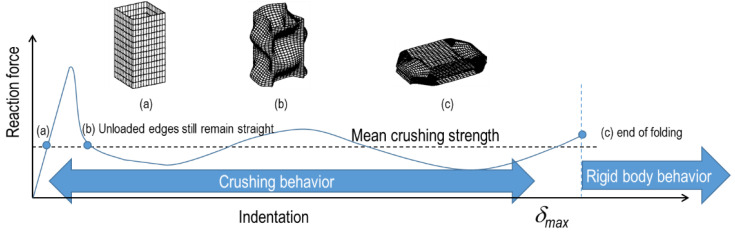
Crushing phase and corresponding reaction force.

**Figure 7 materials-15-02107-f007:**
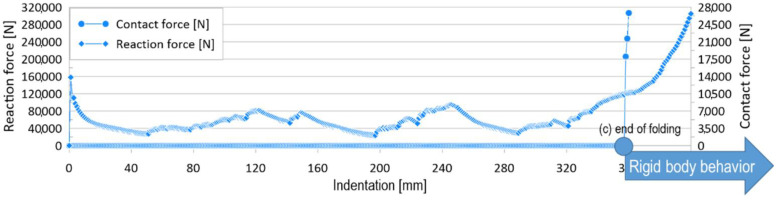
Definition of crushing length.

**Figure 8 materials-15-02107-f008:**
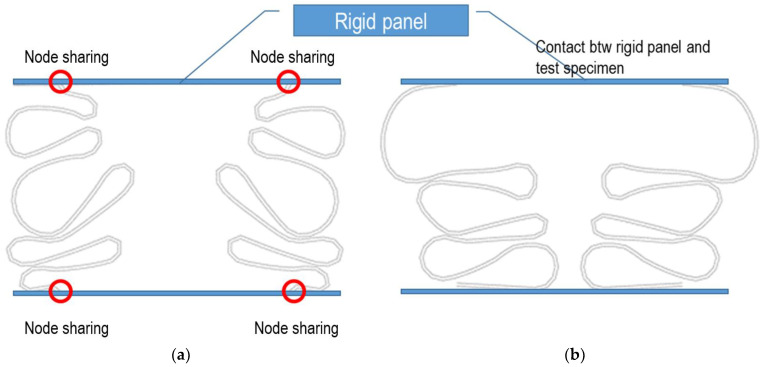
Two boundary conditions applied for the parametric study: (**a**) node sharing and (**b**) contact conditions.

**Figure 9 materials-15-02107-f009:**
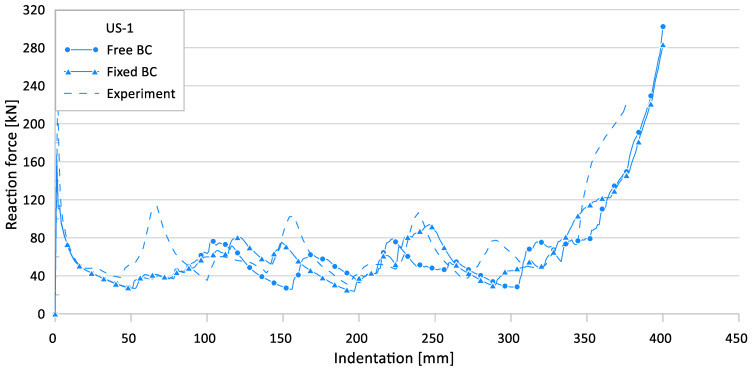
Effect of loading boundary conditions (BC) on crushing behavior of the reaction force for US-1.

**Figure 10 materials-15-02107-f010:**
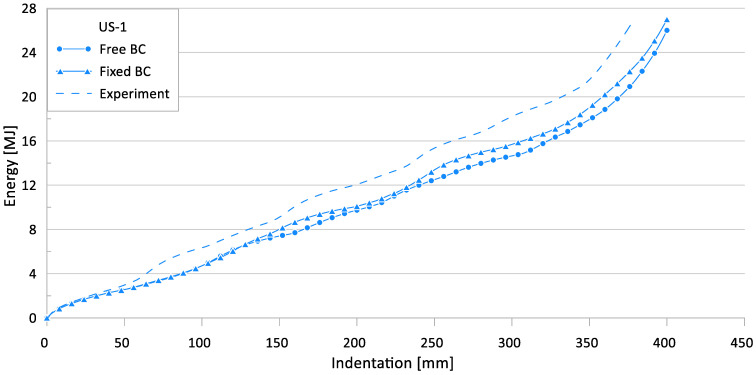
Effect of loading boundary conditions on crushing behavior of the internal energy for US-1.

**Figure 11 materials-15-02107-f011:**
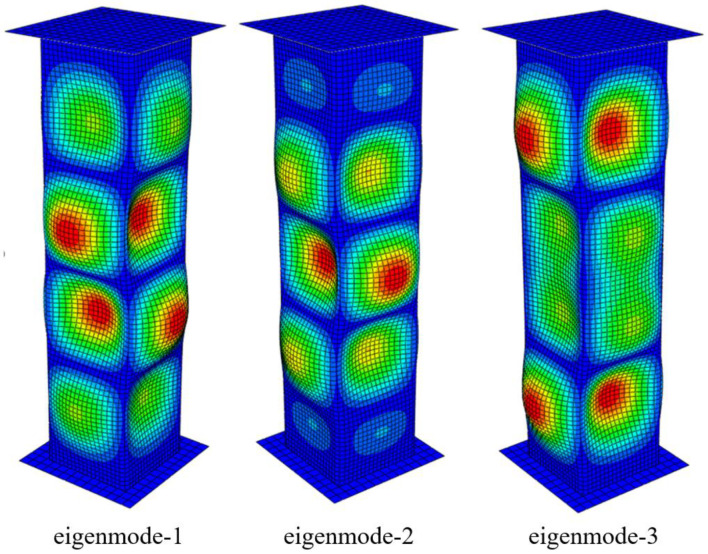
Eigenmodes of the test specimen, obtained from linear buckling finite element assessment.

**Figure 12 materials-15-02107-f012:**
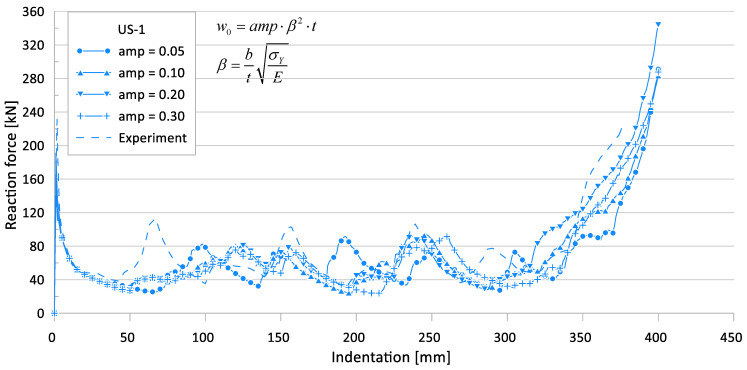
Effect of initial imperfections on crushing behavior of the reaction force for US-1.

**Figure 13 materials-15-02107-f013:**
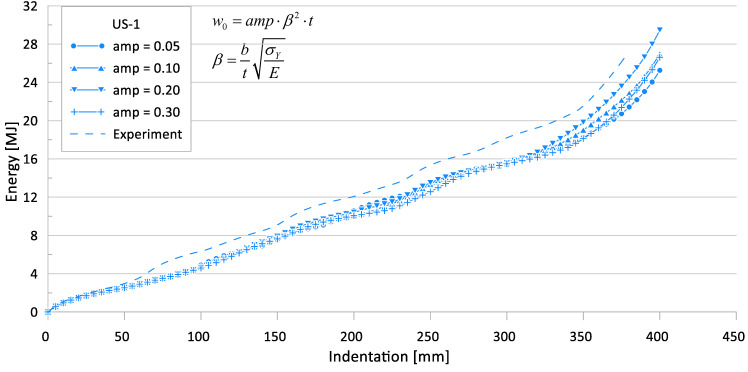
Effect of initial imperfections on crushing behavior of the internal energy for US-1.

**Figure 14 materials-15-02107-f014:**
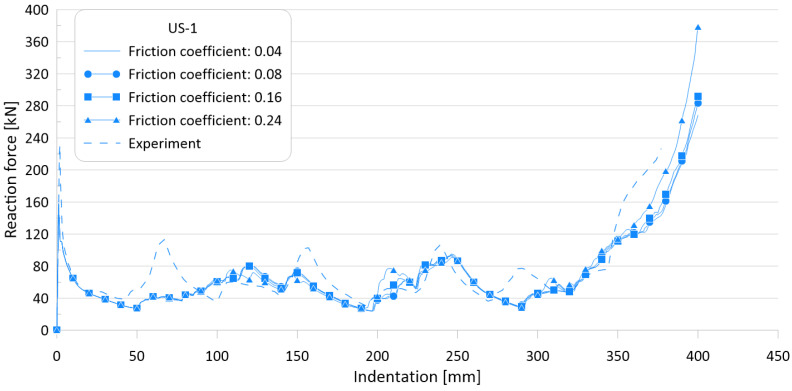
Effect of friction coefficients on crushing behavior of the reaction force for US-1.

**Figure 15 materials-15-02107-f015:**
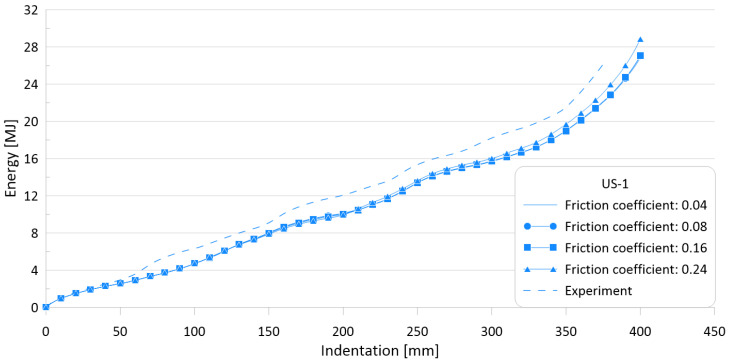
Effect of friction coefficients on crushing behavior of the internal energy for US-1.

**Figure 16 materials-15-02107-f016:**
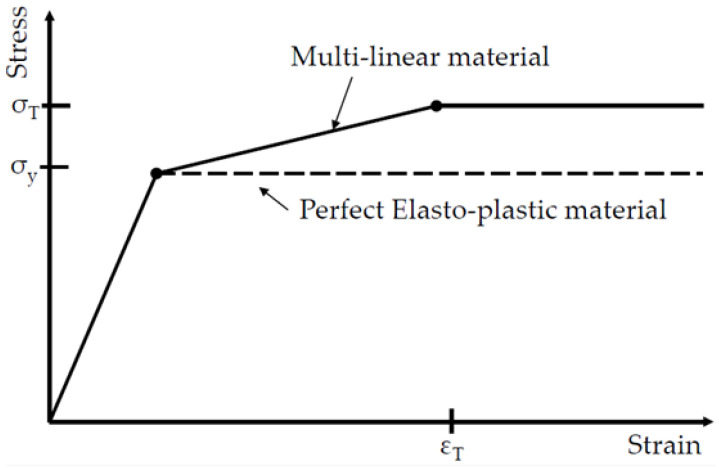
Material property models for nonlinear finite element simulations.

**Figure 17 materials-15-02107-f017:**
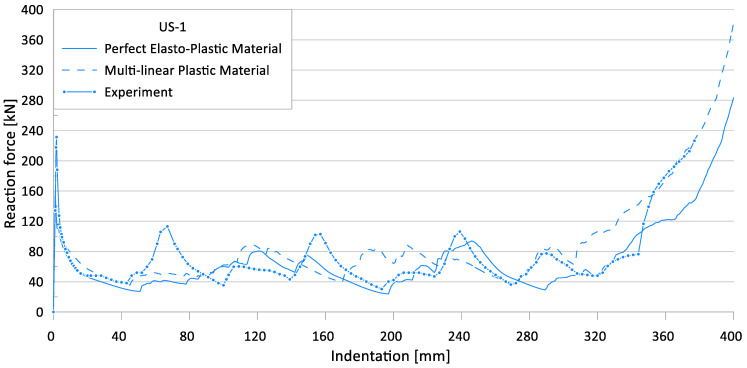
Effect of material models on crushing behavior of the reaction force for US-1.

**Figure 18 materials-15-02107-f018:**
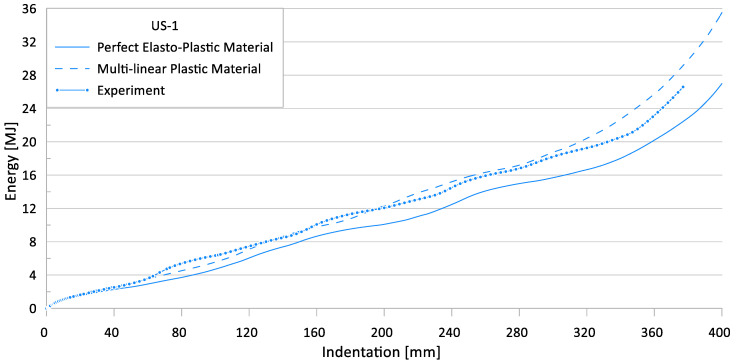
Effect of material models on crushing behavior of the internal energy for US-1.

**Figure 19 materials-15-02107-f019:**
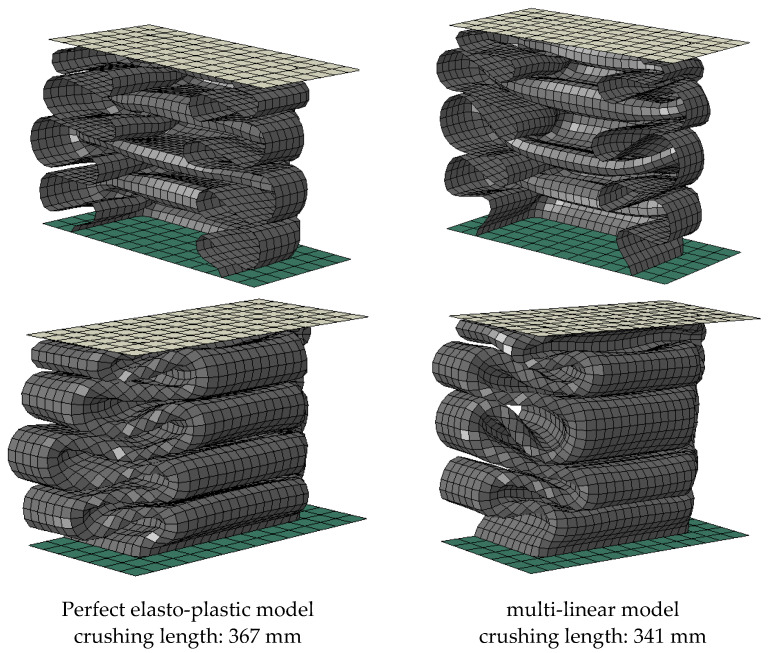
Comparison of deformed shape and crushing length between perfect elasto-plastic and multi-linear models.

**Figure 20 materials-15-02107-f020:**
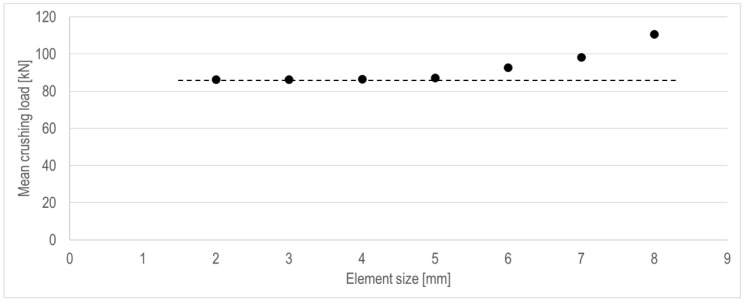
Mean crushing loads, depending on element size of each finite element model.

**Figure 21 materials-15-02107-f021:**
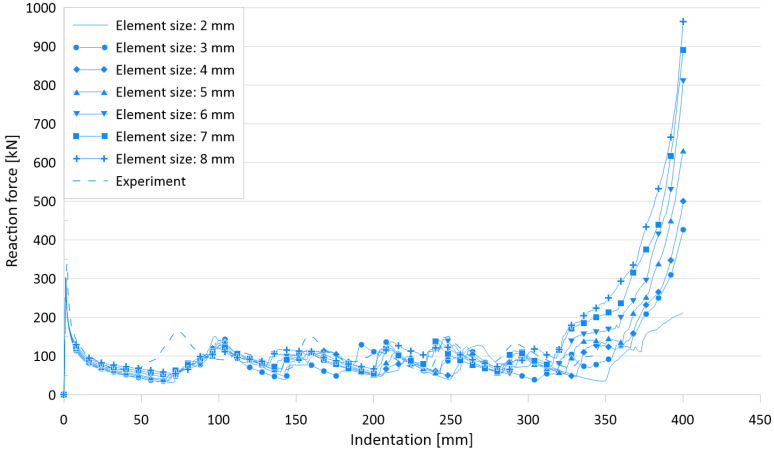
Effect of element sizes on crushing behavior of the reaction force for US-2.

**Figure 22 materials-15-02107-f022:**
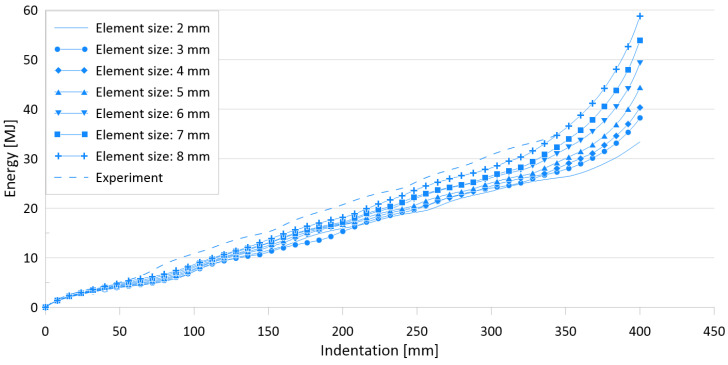
Effect of element sizes on crushing behavior of the internal energy for US-2.

**Figure 23 materials-15-02107-f023:**
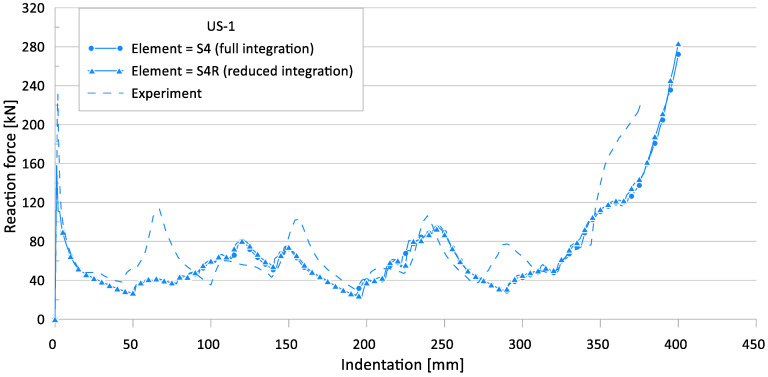
Effect of element type on crushing behavior of the reaction force for US-1.

**Figure 24 materials-15-02107-f024:**
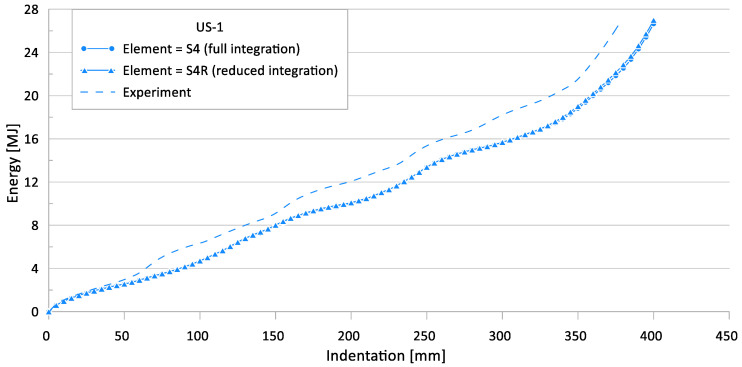
Effect of element type on crushing behavior of the internal energy for US-1.

**Figure 25 materials-15-02107-f025:**
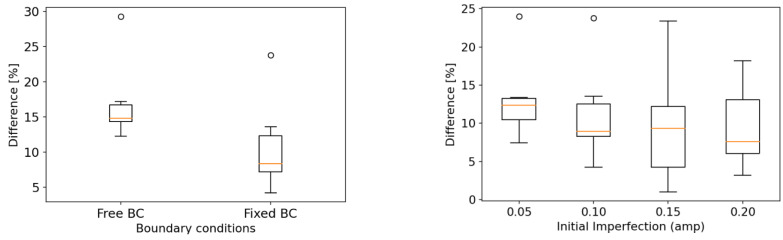
Box-and-whisker plot with the probability density function of a normal distribution.

**Figure 26 materials-15-02107-f026:**
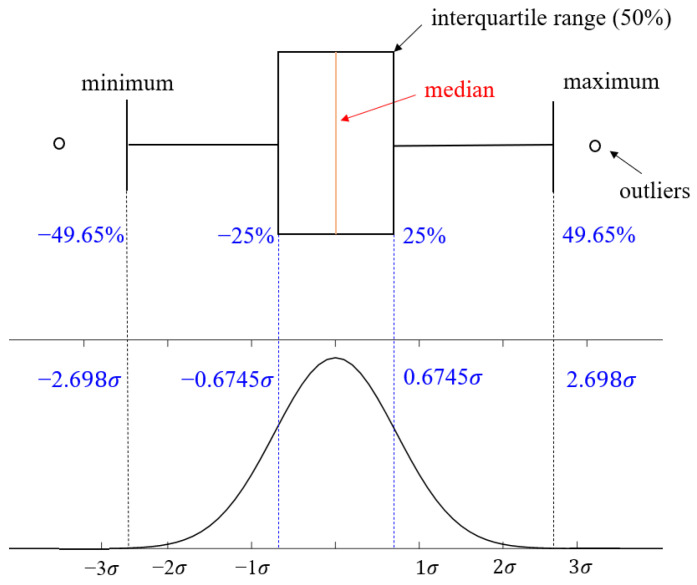
Box-and-whisker plots for the results of the parametric study.

**Figure 27 materials-15-02107-f027:**
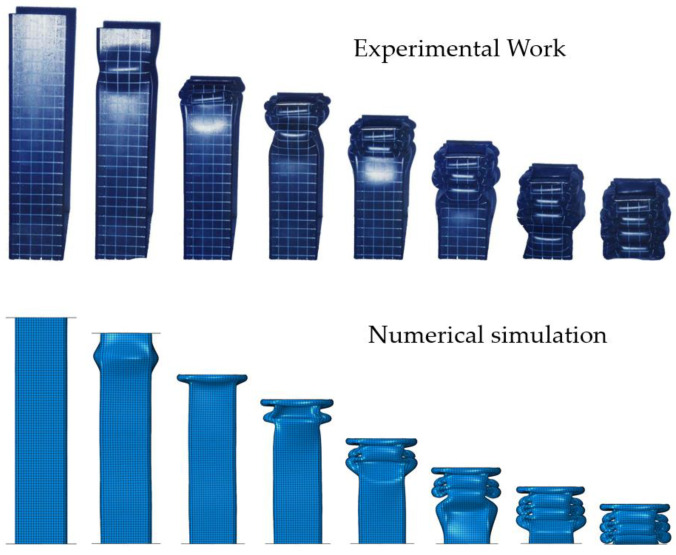
Comparison of the experimental and simulated deformation of the square tube, according to the loading stage (US-1).

**Figure 28 materials-15-02107-f028:**
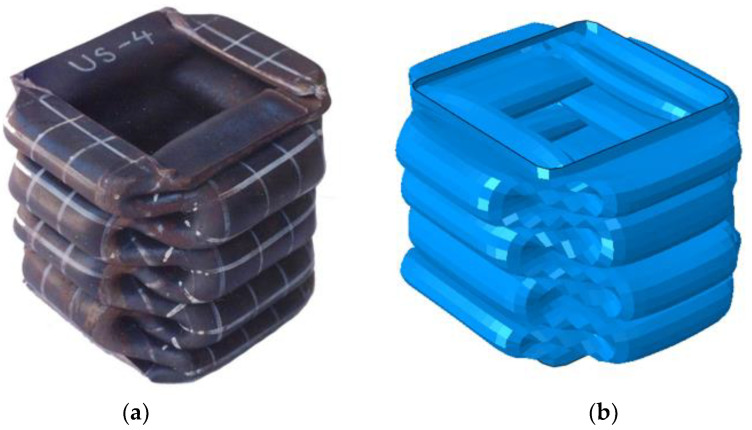
Comparison of the final deformation between the experiment (**a**) and numerical simulation (**b**) (US-4).

**Figure 29 materials-15-02107-f029:**
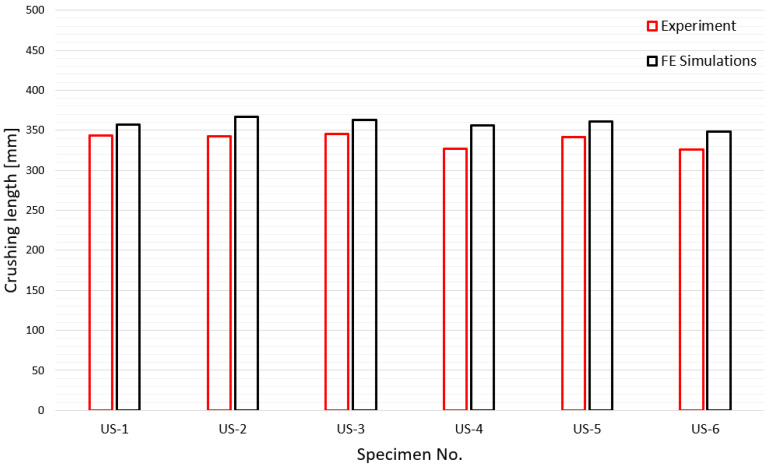
Comparison of experimental and simulated crushing lengths.

**Figure 30 materials-15-02107-f030:**
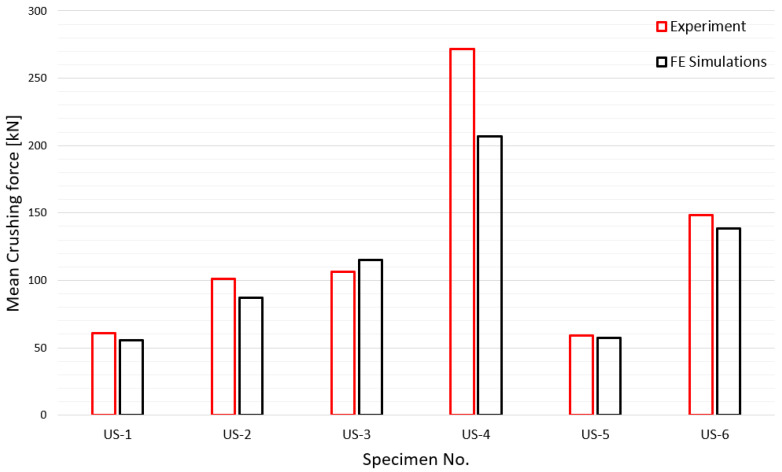
Comparison of experimental and simulated mean crushing forces.

**Table 1 materials-15-02107-t001:** Test specimen dimensions of square tubes under quasi-static loading [29].

Specimen Type	Specimen ID	L ^1^ [mm]	b ^1^ [mm]	t ^1^ [mm]
Unstiffened specimen	US-1	450	100	2.2
	US-2		100	2.8
	US-3		100	3.0
	US-4		100	4.2
	US-5		75	2.1
	US-6		75	3.2

^1^ L = height of square tube, b = section breadth of square tube, t = thickness of square tube.

**Table 2 materials-15-02107-t002:** Mechanical properties of target specimens [29].

Specimen ID	Elongation [%]	E [GPa]	σ_y_ [MPa] (0.2% Proof Stress)	σ_T_ [MPa]	ε_T_
US-1	41.9	199.2	286.1	351.9	0.210
US-2	35.4	205.8	310.8	363.3	0.177
US-3	35.6	207.8	299.3	351.2	0.178
US-4	36.5	199.9	366.8	450.8	0.183
US-5	37.4	200.1	328.5	399.4	0.187
US-6	24.9	209.7	427.6	477.2	0.125

**Table 3 materials-15-02107-t003:** Specification of the finite element analysis.

Item		Note
Finite element solver	ABAQUS standard ABAQUS explicit	For eigenvalue buckling assessment For nonlinear crushing simulation
Material type	Homogeneous isotropic material	
Boundary condition	Fixed type Simply supported type	
Surface contact	Kinematic hard contact option	
Elastic modulus	205,800 MPa	
Poisson’s ratio	0.3	
Yield strength	From the experimental tests	

**Table 4 materials-15-02107-t004:** Test and finite element simulation results by boundary condition for crushing specimens.

Specimen ID	Experimental Works [kN]		Finite Element Simulations [kN]				Difference [%]			
			Free		Fixed		Free		Fixed	
	P_u_ ^(1)^	P_m_ ^(2)^	P_u_ ^(3)^	P_m_ ^(4)^	P_u_ ^(5)^	P_m_ ^(6)^	(^(1)^ − ^(3)^)/^(1)^	(^(2)^ − ^(4)^)/^(2)^	(^(1)^ − ^(5)^)/^(1)^	(^(2)^ − ^(6)^)/^(2)^
US-1	231.3	60.6	167.5	51.4	157.9	55.6	27.6	15.2	31.7	8.2
US-2	337.1	101.0	305.7	83.7	302.3	87.3	9.3	17.2	10.3	13.6
US-3	361.6	106.2	310.1	90.9	315.3	115.2	14.3	14.4	12.8	8.5
US-4	626.0	271.6	583.5	192.0	588.7	207.0	6.8	29.3	6.0	23.8
US-5	193.1	59.0	175.7	50.6	174.9	57.5	9.0	14.3	9.4	4.2
US-6	404.7	148.4	390.4	130.2	388.4	138.2	3.5	12.3	4.0	6.9

**Table 5 materials-15-02107-t005:** Test and finite element simulation results by initial deflection for crushing specimens.

Specimen ID	Experimental Works		FE Simulations			Difference [%]	
	P_u_ ^(1)^ [kN]	P_m_ ^(2)^ [kN]	amp	P_u_ ^(3)^ [kN]^)^	P_m_ ^(4)^ [kN]	(^(1)^ − ^(3)^)/^(1)^ [%]	(^(2)^ − ^(4)^)/^(2)^ [%]
US-1	231.3	60.6	0.05	191.2	54.5	17	10
			0.10	157.9	55.6	32	8
			0.20	157.2	61.2	32	1
			0.30	144.9	51.6	37	15
US-2	337.1	101.0	0.05	327.3	87.5	3	13
			0.10	302.3	87.3	10	14
			0.20	256.9	88.2	24	13
			0.30	227.7	97.8	32	3
US-3	361.6	106.2	0.05	335.5	118.6	7	12
			0.10	315.3	115.2	13	8
			0.20	264.4	114.5	27	8
			0.30	234.2	113.9	35	7
US-4	626.0	271.6	0.05	593.9	206.3	5	24
			0.10	588.7	207.0	6	24
			0.20	576.4	208.1	8	23
			0.30	546.4	222.1	13	18
US-5	193.1	59.0	0.05	194.2	51.3	1	13
			0.10	174.9	57.5	9	4
			0.20	142.7	52.6	26	11
			0.30	134.5	54.3	30	8
US-6	404.7	148.4	0.05	398.0	140.3	2	5
			0.10	388.4	138.2	4	7
			0.20	364.7	145.1	10	2
			0.30	339.6	142.3	16	4

**Table 6 materials-15-02107-t006:** Test and finite element simulation results by friction coefficient for the crushing specimens.

Specimen ID	Experimental Works		FE Simulations			Difference	
	P_u_ ^(1)^ [kN]	P_m_ ^(2)^ [kN]	Friction Coefficient	P_u_ ^(3)^ [kN]	P_m_ ^(4)^ [kN]	(^(1)^ − ^(3)^)/^(1)^ [%]	(^(2)^ − ^(4)^)/^(2)^ [%]
US-1	231.3	60.6	0.04	157.9	55.6	31.7	8.2
			0.08	157.9	55.6	31.7	8.2
			0.16	157.9	55.3	31.7	8.7
			0.24	157.9	54.6	31.7	9.8
US-2	337.1	101.0	0.04	302.3	88.7	10.3	12.2
			0.08	302.3	87.3	10.3	13.6
			0.16	302.3	87.9	10.3	12.9
			0.24	302.3	87.4	10.3	13.5
US-3	361.6	106.2	0.04	343.6	118.8	5.0	11.8
			0.08	343.6	115.2	5.0	8.5
			0.16	343.6	112.5	5.0	5.9
			0.24	343.6	114.1	5.0	7.5
US-4	626.0	271.6	0.04	588.7	207.3	6.0	23.7
			0.08	588.7	207.0	6.0	23.8
			0.16	588.7	204.0	6.0	24.9
			0.24	588.7	204.3	6.0	24.8
US-5	193.1	59.0	0.04	174.9	56.2	9.4	4.7
			0.08	174.9	57.5	9.4	4.2
			0.16	174.9	56.4	9.4	4.4
			0.24	174.9	54.5	9.4	7.6
US-6	404.7	148.4	0.04	388.4	138.2	4.0	6.9
			0.08	388.4	138.2	4.0	6.9
			0.16	388.4	137.3	4.0	7.5
			0.24	388.4	137.9	4.0	7.1

**Table 7 materials-15-02107-t007:** Test and finite element simulation results by nonlinear material model.

Specimen No.	P_u_ [kN]			P_m_ [kN]			Difference [%]			
							P_u_		P_m_	
	Exp.	PE	MLM	Exp.	PE	MLM	PE	MLM	PE	MLM
US-1	231.3	157.9	159.5	60.6	55.6	68.7	31.7	31.0	8.3	13.4
US-2	337.1	302.3	304.1	101.0	87.3	100.6	10.3	9.8	13.6	0.4
US-3	361.6	343.6	344.8	106.2	115.2	129.7	5.0	4.6	8.5	22.1
US-4	626.0	588.7	589.6	271.6	207.0	252.6	6.0	5.8	31.7	31.0
US-5	193.1	174.9	176.4	59.0	57.5	64.0	9.4	8.6	2.5	8.5
US-6	404.7	388.4	388.8	148.4	138.2	157.0	4.0	3.9	6.9	5.8

**Table 8 materials-15-02107-t008:** Deformed shape and number of elements for each folding length by element size.

Size	Deformed Shape—Edge Line	Deformed Shape—Half of the Specimen	No. of Elements for Each Folding
2 mm	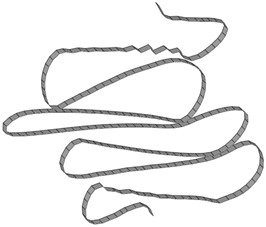	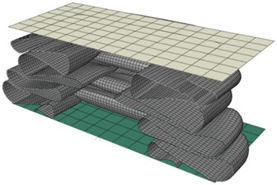	63
3 mm	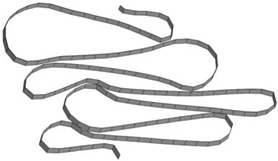	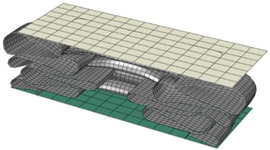	41
4 mm	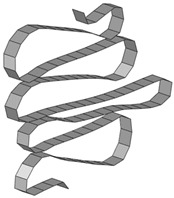	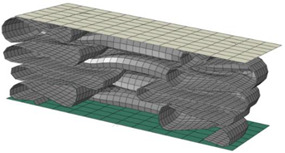	31
5 mm	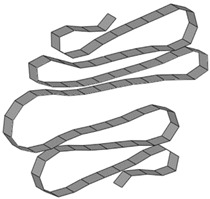	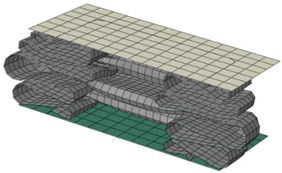	24
6 mm	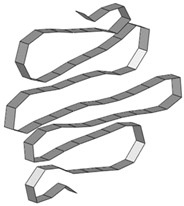	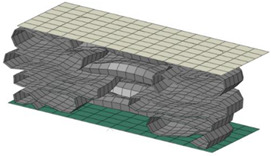	20
7 mm	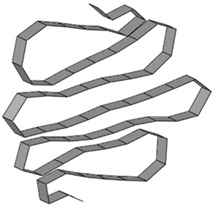	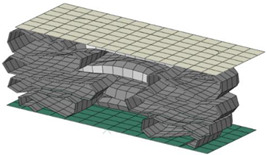	19
8 mm	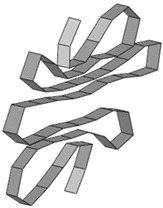	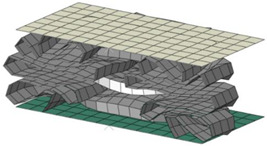	17

**Table 9 materials-15-02107-t009:** Test and finite element simulation results by element size.

Experiment		FE Simulations			Difference	
P_u_ [kN]	P_m_ [kN]	Element Size [mm]	P_u_ [kN]	P_m_ [kN]	P_u_ [%]	P_m_ [%]
337.1	101.1	2	300.2	86.3	10.9	14.6
		3	300.5	86.4	10.9	14.5
		4	301.3	86.5	10.6	14.4
		5	302.3	87.3	10.3	13.6
		6	302.3	92.7	10.3	8.3
		7	301.8	98.4	10.5	2.7
		8	303.7	110.7	9.9	9.5

**Table 10 materials-15-02107-t010:** Test and finite element simulation results by element type.

Specimen No.	Experiment [kN]		FEM [kN]						Difference [%]			
			S4 ^1^		S4R ^1^		(^(3)^ − ^(5)^)/^(3)^	(^(4)^ − ^(6)^)/^(4)^	(^(1)^ − ^(3)^)/^(1)^	(^(2)^ − ^(4)^)/^(2)^	(^(1)^ − ^(^^5^^)^)/^(1)^	(^(2)^ − ^(^^6^^)^)/^(2)^
	P_u_ ^(1)^	P_m_ ^(2)^	P_u_ ^(3)^	P_m_ ^(4)^	P_u_ ^(5)^	P_m_ ^(6)^	P_u_	P_m_	P_u_	P_m_	P_u_	P_m_
US-1	231.3	60.6	157.8	55.9	157.9	55.6	0.06	0.54	31.78	7.76	31.73	8.25
US-2	337.1	101.0	301.9	84.4	302.3	87.3	0.13	3.44	10.44	16.44	10.32	13.56
US-3	361.6	106.2	343.7	112.0	343.6	115.2	0.03	2.86	4.95	5.46	4.98	8.47
US-4	626.0	271.6	588.9	203.2	588.7	207.0	0.03	1.87	5.93	25.18	5.96	23.78
US-5	193.1	59.0	174.9	54.5	174.9	57.5	0.00	5.50	9.43	7.63	9.43	2.54
US-6	404.7	148.4	388.7	135.4	388.4	138.2	0.08	2.07	3.95	8.76	4.03	6.87

^1^ S4 = full integration four-node shell element, S4R = reduced integration four-node shell element.

**Table 11 materials-15-02107-t011:** Summary of the determined parameters for the finite element simulation.

Parameter	Selected Case
Boundary condition	Fix
Initial imperfection	$0.1{\;\beta }^{2}\cdot t$
Friction coefficient	0.08
Material property model	perfect elasto-plastic material
Finite element mesh size	5 mm
Element type	S4R (reduced integration four-node shell element)

## Data Availability

The data presented in this study are available on request from the corresponding author.

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
