# Peer review of "Numerical and Experimental Investigation of Quasi-Static Crushing Behaviors of Steel Tubular Structures"

_materials, 2022, doi:10.3390/ma15062107_

Round 1

Reviewer 1 Report

Numerical and Experimental Investigation of Quasi-static Crushing 
Behaviors of Steel Tubular Structures
Young IL Park, Jin-Seong Cho, and Jeong-Hwan Kim

Comments:

1. Section 2 : Experimental works .........................in this 
chapter ???
2. Why was the crushing speed of 0.05 mm/s chosen ?
3. If authors could use Origin to draw figures, the quality and 
legibility would be much better.
4.The ultimate crushing loads obtained with the four different friction 
coefficients were found to be identical to each other, but the mean 
crushing loads were slightly different. What is the reason behind this 
behavior ?
5.The level of initial imperfection has a particularly large impact on 
the ultimate crushing strength, but it has a relatively smaller effect 
on the mean crushing load. Kindly explain the reason behind this behavior?
6. Any other conditions for welding, joints kept in parameters while 
doing the simulations ?

Overall, a good paper.

Reviewer 2 Report

According to my opinion, the manuscript represents good research in the field of crushing behavior of the tubular structures. The idea, as well as experimental and numerical setup, are appropriate. After the very detailed analysis of the many possible influential factors (6 in this case), the final numerical model is defined and the obtained results show that it can be considered a reliable one.

I recommend the manuscript to be accepted for publishing and have only a few minor remarks:

  1. On page 3 you explain the experimental setup but you do not mention any international standard according to which the testing is defined, is there any?
  2. On page 4, in table 2 you give the mechanical properties of target specimens but I could not see the designation of the used steel. Please add it in the text.
  3. On page 11 you involve different friction coefficients in order to determine their influence on the crushing process. Have you considered the case in which you exclude the friction completely? For example, use graphite-based lubricant?
  4. On page 23 I find unnecessary rows 437-442 since you mention all the parameters in table 11 on page 21. I suggest deleting them.

Reviewer 3 Report

  • Please add the novelty of the work at the end of Introduction section.
  • Please modify the ABSTRACT with the incorporation of methodology component.
  • In Table 9, the repetition of mm, % etc. may be avoided by incorporating Element size (mm) and Pu (%) and Pm (%) respectively.
  • Page 22, lines 417 to 419, please give the literature support for the statement “ --- because the ability to absorb energy is the most important 418 design factor for a tube subjected to axial crushing.”
  • Experimental crushing length and force considered in Figures 28 and 29, please mention the number of trials considered for the experimental data.

Reviewer 4 Report

The paper is devoted to numerical simulation of crushing behavior of steel tubular structures. The topic is relevant and fits within the scope of journal Materials. The paper contributes to the development of methods for prediction of quasi-static fracture of spatial structures. Experimental raw data of [30] were used in numerical research on the influence of determining factors on accuracy and efficiency of numerical results.

  • Description of multi-linear plasticity model is missing. Typical stress-strain curve in Figure 14 contains descending softening branch, which cannot be modeled utilizing piecewise linear hardening model. Which maximum magnitude of equivalent plastic strain was obtained in simulation of tubular structures? Whether it exceeded the value of strain-to-fracture of constituent material obtained in tensile test?
  • Note that the simulation results were compared with the experimental data obtained for square pipes having the same length (450 mm), similar cross-sectional dimensions and wall thicknesses. At the same time, the authors indicated that the main goal of the study was to indicate an efficient and accurate numerical modeling method for the analysis of the impact resistance of ship structures and offshore industrial applications. The possibility of using the results of the study on small-sized model samples to obtain adequate predictions of the impact resistance of ship structures and offshore industrial applications should be justified.
  • Authors’ arguments regarding the use of an ideal elastic-plastic model for predicting the shape of deformation and the average compressive force of tubular structures should be stated in more detail.
  • Table 3. Whether the explicit solver was used in this study?

Reviewer 5 Report

The introduction clearly sets out the problem under investigation.

The methodology is adequate. This explains how the data was collected.

The procedures are ordered in a logical way.

The method is explained in detail.

The results are clearly set out in a logical sequence.

The conclusions are supported by the results.

Author Response

Thank you very much for your review.